# A Characterization of Brain Area Activation in Orienteers with Different Map-Recognition Memory Ability Task Levels—Based on fNIRS Evidence

**DOI:** 10.3390/brainsci12111561

**Published:** 2022-11-17

**Authors:** Yang Liu, Su Lu, Jingru Liu, Mingsheng Zhao, Yue Chao, Pengyang Kang

**Affiliations:** 1School of Physical Education, Shaanxi Normal University, Xi’an 710062, China; 2School of Physical Education, Ankang University, Ankang 725000, China; 3Sports Department, Xi’an University of Posts and Telecommunications, Xi’an 710121, China; 4School of Foreign Languages, Shaanxi Normal University, Xi’an 710062, China

**Keywords:** orienteering, functional near-infrared spectroscopy, prefrontal lobe (PFC), map recognition

## Abstract

Background: Mapping memory ability is highly correlated with an orienteer’s level, and spatial memory tasks of different difficulties can reveal the spatial cognitive characteristics of high-level athletes. Methods: An “expert–novice” experimental paradigm was used to monitor behavioral performance and changes in cerebral blood oxygen concentration in orienteering athletes with tasks of different difficulty and cognitive load using functional near-infrared spectroscopic imaging (fNIRS). Results: (1) there was no difference between high-/low-level athletes’ map recognition and memory abilities in the non-orienteering scenario; (2) with increasing task difficulty, both high-/low-level athletes showed significantly decreasing behavioral performance, reduced correctness, longer reaction time, and strengthened cerebral blood oxygen activation concentration. There was no significant difference in L-DLPFC cerebral oxygen concentration between high-/low-level athletes in the simple map task, and the cerebral oxygen concentration in all brain regions was lower in the expert group than in the novice group in the rest of the task difficulty levels; (3) the correctness rate in the expert group in the complex task was closely related to the activation of the right hemisphere (R-DLPFC, R-VLPFC). Conclusions: Experts have a specific cognitive advantage in map-recognition memory, showing higher task performance and lower cerebral blood oxygen activation; cognitive load constrains map-recognition memory-specific ability and produces different performance effects and brain activation changes on spatial memory processing.

## 1. Introduction

Orienteering is a sport in which participants reach each point marker on a map with the help of a compass and a map, and the winner is the one who takes the least time [1]. In the course of the competition, orienteers obtain effective information from the map, process and encode it, and continuously extract and convert the map information into the real environment information of the real scene to find the target object [2]; this requires athletes to have good spatial memory ability for map information in the process of figure and scene switching [3], mainly including athletes’ effective recognition of various features, landform symbols, and path routes on the map. This requires athletes to have good spatial memory for map information in the process of switching between map and scenery [3], which mainly includes athletes’ effective recognition of various feature and landscape symbols and route information on the map and correct matching of map information in the search process in the field environment [4].

Spatial working memory is the temporary preservation of visual–spatial information, and the cognitive system saves the corresponding information in different forms according to the specific characteristics of the information to be processed, and then further refines the stored information and applies the processed information to various complex cognitive activities [5]. As a type of memory, spatial memory is the storage and processing of spatial information [6]. Currently, studies on the cognitive abilities of orienteers have confirmed that orienteers have good map-recognition memory [7], and memory ability is highly correlated with athlete level, as in the study by Lunze, who found that the map-recognition ability of German national orienteering team athletes was higher than that of ordinary university students, and that the better spatial memory ability of professional athletes comes from years of sports experience and expertise training [8]. However, further research is needed to clarify whether this memory advantage of high-level orienteers is innately developed or acquired through later training. Previous studies comparing the visual memory of novice and veteran athletes and visual memory representations of pilots [9,10,11] have found that experience and specialized practice cause individuals to develop cognitive adaptations and their own memory strategies, resulting in changes in memory breadth, forms of expertise organization, and the ability to acquire knowledge from memory.

In a related study on orienteering, Liu Yang [12] et al. found that difficulty load (dot conformation and color symbols in orienteering maps) influenced the processing efficiency of visual attention search and route decision-making of orienteers when identifying orienteering maps to extract valid information. Zhu Yu et al. [13] used the eye-tracking technique and found that under different conditions of task difficulty, orienteers differed in terms of gaze duration, number of gazes, and amplitude of eye jumps. In a study on the decision characteristics and training interventions of badminton players in different difficulty tasks, Bo Jianmei [14] found that task difficulty was an important variable restricting the cognitive performance of athletes. Therefore, map difficulty was used as an in-group variable in this study to examine the behavioral performance, cerebral blood oxygen characteristics, and spatial memory under different task difficulty conditions in map-literate orienteering athletes of different levels.

The existing research paradigms in orienteering are mostly behavioral indicators (map reading skills, route decision-making skills, etc.), exploration, and empirical descriptions, lacking in-depth scientific exploration. In recent years, with the continuous development of cognitive neuroscience, fNIRS (functional near-infrared spectroscopy) has been more and more widely applied in the field of sports, with certain progress and breakthroughs in basketball [15], tai chi [16], table tennis [17], shooting sports [18], and aerobic sports [19]. The fNIRS has the advantages of being mobile, noiseless, easy to operate and maintain, rarely limited by subjects, and applicable to all possible participating populations. The prefrontal lobe (PFC) was found to play an important role in cognitive processes by the fNIRS technique. It has been suggested that spatial tasks during memory route and path planning are related to structures such as the PFC [20]. The PFC is the area that receives processed external information from other functional areas of the brain and then integrates brain information such as memory and intention to immediately make a rational plan [21,22,23]. The dorsolateral prefrontal lobe (DLPFC) and ventral lateral prefrontal lobe (VLPEC), as the PFC’s main functional areas, play an important role in brain functions related to motor cognition and are important for contextual planning and decision-making. Indeed, many functional neuroimaging studies have reported activation of the PFC during the memory retrieval process [24,25,26]. Therefore, the present study selected the prefrontal lobe as the area of interest to further investigate whether different levels of orienteering athletes’ spatial memory behavioral performance for map recognition corresponds to exhibiting different brain activation characteristics. What kinds of brain activation characteristics do orienteers show for spatial memory tasks under different cognitive loads?

To address the above questions, the study proposes the following hypotheses: (1) There are no spatial memory differences between expert and novice athletes in non-specific scenarios. (2) During the map recognition and memory task, expert and novice groups show different behavioral performance, and expert athletes will show a specialized cognitive advantage. (3) Different map difficulties will cause differences in Oxy-Hb (deoxyhemoglobin) signals in PFC functional brain regions during the map recognition and spatial memory task, and there are differences in brain activation levels of athletes at different levels. This study can provide theoretical support and practical guidance for scientific research on orienteering sports programs to achieve the optimization of training.

## 2. Materials and Methods

### 2.1. Participants

In this study, 15 subjects from an orienteering expert group and 15 from a novice group were selected to voluntarily enroll in this experiment. The expert group was made up of professional players from the Chinese orienteering team with more than 4 years of training, while the novice group comprised orienteering practitioners from a Chinese university.

Before the test, the basic information of all subjects, such as height, weight, age, and BMI, was recorded(see Table 1). The subjects were informed of the corresponding requirements of the experiment (schedule and precautions, etc.) one day in advance, and were asked to ensure sufficient sleep before the experiment, as well as head cleanliness and no strenuous exercise during the day before the experiment. Consent was obtained from the subjects, they signed an informed consent form for the experiment, and they will receive the appropriate payment after completing the experiment. The study was approved by the ethics committee of Shaanxi Normal University.

### 2.2. Experimental Design

The experiment consisted of two spatial memory task scenarios: a non-specific task scenario and a specific task scenario. The Corsi block-tapping task paradigm, which is widely used in clinical and experimental studies to assess visuospatial attention, working memory processes, and spatial memory breadth in healthy participants and patients with known or suspected brain injury, was chosen. It requires participants to remember the sequential order of blocks in a sequence and to recreate a series of scenes by tapping on the blocks, allowing measurement of visuospatial short-term memory capacity [27]. Differences in the general spatial memory ability of orienteers at different levels were assessed by recording the subjects’ spatial memory breadth scores.

The specialized scene memory task set two task difficulties, simple map and complex map, with a 2 (athlete level: expert group, novice group) × 2 (map difficulty: simple map, complex map) two-factor mixed experimental design. Athlete level was the between-group variable and map difficulty was the within-group variable to examine the differences in spatial memory abilities of orienteers of different levels in orienteering sport-specific scenarios with different task difficulties. The experimental tasks were presented on the computer side, and the behavioral data of the subjects were recorded by E-prime 2.0 (Psychology Software Tools Inc, Sharpsburg, PA, USA). The dependent variables were the correctness rate, reaction time, and Oxy-Hb blood oxygen concentration of the subjects.

### 2.3. Experimental Materials

Specialized scene stimulation materials: 800 × 600 pixel orienteering standard competition maps were used, which were made by three nationally qualified orienteering mapmakers and rated for difficulty, to establish the two difficulty levels of simple maps and complex maps. Simple maps are mostly town scenes, and map information is mostly about buildings and other features of landforms (see Figure 1a). Complex maps are mostly mountain scenes, and map information is mostly about mountains and other geomorphic features (see Figure 1b). All stimulus materials were drawn using orienteering maps that had not been featured in orienteering events in the recent year, using the OCAD 11.0 version mapping software developed by the International Orienteering Federation (Karlstad, Sweden).

### 2.4. fNIRS Test Protocol

The experimental instrument was a portable near-infrared spectral brain function imaging system, Nirsport 2, to detect the hemodynamic signal of local brain regions during the subject’s task. Using the international 10–20 localization system as a reference, the lowest probe was placed along the Fp1–Fp2 line, and the system’s own PFC template was used, including 13 light source probes as well as 8 receiver probes, which together constituted 28 measurement channels, with the sampling frequency set to 7.8125 Hz (see Figure 2). An elastic head cap was used to fix the template and the head. When placing the detector probes and NIR light source probes in the template, the subject’s hair needed to be fully ruffled to ensure full contact between the probes and the scalp. The 3D digitizer (FASTRAK system) was used to locate the probe position, and the NIRS_SPM software determined the MNI spatial coordinates of the fNIRS channel position by the probabilistic alignment method to find the corresponding brain regions in the adult Brodmann area (Brodmann) atlas, and a total of four ROIs were delineated (see Table 2)—left brain regions: left ventral lateral prefrontal cortical region (L-VLPFC), left dorsolateral prefrontal cortical area (L-DLPFC); right brain area: right ventral lateral prefrontal cortical area (R-VLPFC), right dorsolateral prefrontal cortical area (R-DLPFC) (see Table 2).

### 2.5. Experiment Procedures

#### 2.5.1. Non-Orienteering Scenario Experiment Flow

All experiments were conducted in the psychological laboratory of Shaanxi Normal University, and the test program was written using “E-prime 2.0”, which was designed to respond to the spatial working memory breadth ability with the correct rate of acquisition. The experiment consisted of two phases: a practice phase and a formal test phase. The first phase consisted of an “instructional statement”, which aimed to let the subjects understand the procedure of the experiment. After the blinking is over, please “immediately remember the square that just blinked in order”, then use the mouse to “click on the three positions” that just blinked in order to answer, and press the “space” button to continue to the next test (see Figure 3 for the flow chart). If the subject gets more than 75% correct twice in a row (if correct only once, the score will be increased by 0.5 points), the number of blinking points will be increased by one, and the number of blinking points corresponds to the score, for example, if the final number of blinking points is 4, the score will be 4.

#### 2.5.2. Orienteering Scenario Experiment Flow

The experimental procedure was written by E-prime 2.0 software. Before the experiment, the subjects were asked to fill in the basic situation information collection form, informing them of the purpose of the experiment as well as the procedure, and briefly introducing the basic principle of fNIRS and the precautions taken to eliminate their nervousness and anxiety. After that, the subjects were asked to put on the fNIRS photopolar cap and start the experiment. All stimuli were presented on a computer screen (see Figure 4).

The experiment consisted of two phases: a practice phase and a formal test phase. In both phases, each trajectory was processed identically, with the purpose of the practice phase being to help participants familiarize themselves with the experiment; the stimulus materials used in this phase would not appear in the subsequent formal test, and no data were collected during this period. The formal test could not begin until the subjects’ blood oxygen data were collected in the quiet state at the end of the practice session.

The subjects were first asked to familiarize themselves with the experimental instructions, then a 5000-ms original map stimulus memory phase appeared, in which the subjects were asked to memorize the information points and path information as quickly and accurately as possible, followed by a 13,000-ms option map phase on the screen (in which the option map consisted of three alternative items, located on the upper middle, lower left, and lower right sides of the screen, corresponding to the response keys “W”, “A”, and “D”), the subject selected the picture that corresponded to the memory stimulus as accurately and quickly as possible, and pressed the response key corresponding to it. The system automatically recorded the correctness rate and response time of the subject, and then let the subject stay relaxed for 2000 ms until the next trial.

A total of 4 practice trials and 50 experimental trials were presented to the subjects, of which 25 experimental trials included 25 simple maps and 25 complex maps. The system presented the two types of maps in order of difficulty (the complex maps were presented after the simple maps), and the behavioral data and fNIRS data of the subjects on the task were recorded in a file; the process was cyclic until the task was completed (see Figure 3).

### 2.6. Data Processing and Statistics 

#### 2.6.1. Behavioral Data Processing 

Subjects were required to make judgments about the stimuli in the shortest possible time during the test, and their response times and correctness rates were recorded during the test. In order to reduce the unnecessary influence of extreme values on the results, the extreme values with large gaps were removed, and the data outside the range of mean ± 3 standard deviations were also removed and did not enter the subsequent statistical analysis. With the help of SPSS 25.0 software (SPSS Inc., Chicago, IL, USA), a normal distribution test was performed, and a threshold value greater than 0.05 indicated that a normal distribution was obeyed. Independent samples *t*-tests were conducted for subjects’ spatial memory breadth scores in non-specific scenarios, and two-factor ANOVAs of group (expert/novice) × map difficulty (simple map/complex map) were performed for correctness and response time in specific scenarios; if there was an interaction, Bonferroni’s method was used for multiple comparison correction, with the significance level set at *p* < 0.05.

#### 2.6.2. fNIRS Data Processing

The Nirsport2 system can solve the optical data collected by Lambert–Beer’s law to obtain the Oxy-Hb, Deoxy-Hb, and Total-Hb signals [28]. The Oxy-Hb signal was used to examine the level of brain changes in subjects because it is more realistic and effective than Deoxy-Hb in reflecting the level of neural activation in the brain [29] In this study, a band-pass filter was used (components greater than 0.1 Hz and less than 0.01 Hz were filtered out) to filter out the effects of heartbeat and respiration from the fNIRS data, and principal components analysis (PCA) was used to remove motion artifacts [30]. Oxy-Hb values were then averaged across all trials in each task condition, and the mean value of each sampling point per channel per unit time (1 s before to 5 s after the start of the trial) for the subject in each task condition was obtained; the oxy-Hb data for channels 6–7 included in the ROIs (regions of interest) were averaged, and this mean value was the blood oxygen signal for that ROI. For the same behavioral data, a two-factor repeated measures ANOVA was performed on the Oxy-Hb data for each index with the help of SPSS 25.0 software, corrected using the Greenhouse–Geisser method. If an interaction occurred it was corrected using the Bonferroni method for further post hoc analysis, with the significance level set at *p* < 0.05.

## 3. Results

### 3.1. Results of Spatial Memory Ability for Non-Orienteering Scenes

Using the form of mean ± standard deviation (M ± SD), Table 3 presents the mean and dispersion of spatial memory breadth scores for novices and experts in the spatial memory breadth task in the undirected scenario, respectively. The results revealed that there was no significant difference in spatial working memory breadth between the expert and novice groups [t = 6.43, *p* > 0.05].

### 3.2. Results of Spatial Memory Ability for Orienteering Scenes

A two-factor ANOVA of 2 (athlete level: expert group, novice group) × 2 (map difficulty: simple map, complex map) was used to statistically analyze the behavioral data on correctness and response time in the spatial memory task (see Table 4).

Correctness results: the ANOVA results showed a significant main effect of movement level (F = 13.618, *p* < 0.001, η2 = 0.532), with the expert group significantly higher than the novice group; a significant main effect of map difficulty type (F = 35.171, *p* = 0.000, η2 = 0.760), with simple maps significantly higher than complex maps; and a map difficulty type × athlete level interaction effect that was not significant (F = 0.147, *p* = 0.708, η2 = 0.746).

Results at reaction: ANOVA results showed a significant main effect of movement level (F = 17.311, *p* = 0.001, η2 = 0.591), which was significantly lower in the expert group than in the novice group; a significant main effect of map difficulty type (F = 4.726, *p* = 0.050, η2 = 0.283),with a significantly lower simple map than a complex map; and an interaction effect of map difficulty type × athlete level effect that was not significant (F = 0.386, *p* = 0.546, η2 = 0.031).

### 3.3. Results of fNIRS Data for the Orienteering Scene Spatial Memory Task

A 2 (athlete level: expert group, novice group) × 2 (map difficulty: simple map, complex map) two-factor repeated measures ANOVA was used to explore the Oxy-Hb concentrations and activation patterns in the left and right ventral lateral prefrontal lobes (L-VLPFC, R-VLPFC) and the left and right ventral medial prefrontal lobes (L-DLPFC, R-DLPFC) of subjects during the spatial memory task (see Table 4, Figure 5 and Figure 6). As shown in Table 5, Figure 5 and Figure 6, there was no remarkable difference in cerebral blood oxygen concentration in L-DLPFC brain regions between the expert and novice groups only in the simple map condition; all other brain regions showed less cerebral blood oxygen activation in the expert groups than in the novice group in both map conditions; and cerebral blood oxygen activation in all brain regions increased in both expert and novice groups as the map difficulty increased.

#### 3.3.1. Left Ventral Lateral Prefrontal Lobe (L-VLPFC)

The results showed a significant main effect of motor level (F = 4.754, *p* = 0.050, η2 = 0.284), with less cerebral oxygen activation in the expert group than in the novice group; a significant main effect of map difficulty (F = 6.142, *p* = 0.029, η2 = 0.339), with higher cerebral oxygen activation in the complex map task condition than in the simple map. The map difficulty × motor level interaction effect was not significant (F = 0.130, *p* = 0.725, η2 = 0.011).

#### 3.3.2. Left Dorsolateral Prefrontal Lobe (L-DLPFC)

The results showed a non-significant main effect of motor level (F = 3.815, *p* = 0.074, η2 = 0.241), a significant main effect of map difficulty (F = 18.752, *p* = 0.001, η2 = 0.610), and higher cerebral blood oxygen activation in the complex map task condition than in the simple map. The map difficulty × motor level interaction effect was significant (F = 16.692, *p* = 0.002, η2 = 0.582).

Further simple effects analysis revealed a non-significant main effect of motor level on the simple map (F = 0.310, *p* = 0.588, η2 = 0.025); on the complex map, the main effect of motor level was significant (F = 9.590, *p* = 0.009, η2 = 0.444) and cerebral oxygen activation in the expert group was lower than in the novice group.

#### 3.3.3. Right Ventral Lateral Prefrontal (R-VLPFC)

The results showed a significant main effect of motor level (F = 5.877, *p* = 0.032, η2 = 0.329), with less cerebral oxygen activation in the expert group than in the novice group; a significant main effect of map difficulty (F = 8.629, *p* = 0.012, η2 = 0.418), with higher cerebral oxygen activation in the complex map type condition than in the simple map; and a non-significant interaction effect of map difficulty × motor level (F = 1.998, *p* = 0.183, η2 = 0.143).

#### 3.3.4. Right Dorsolateral Prefrontal Lobe (R-DLPFC)

The results showed a significant main effect of motor level (F = 13.091, *p* = 0.004, η2 = 0.522), with less cerebral oxygen activation in the expert group than in the novice group; a significant main effect of map difficulty (F = 22.725, *p* = 0.000, η2 = 0.654), with higher cerebral oxygen activation in the complex map type condition than in the simple map; the interaction effect of map difficulty × motor level was not significant (F = 0.060, *p* = 0.810, η2 = 0.005).

### 3.4. Correlation Analysis of the Correctness of Spatial Memory Tasks with Different Map Difficulty and the Intensity of Activation of Brain Interest Areas

The correlation analysis between Oxy-Hb concentrations and behavior (correct rate) for each ROI at different levels to athletes reperforming the spatial memory task was conducted to explore the degree of correlation between activation intensity and behavioral performance, as shown in Table 6 and Figure 7.

As shown in Figure 7, the cerebral blood oxygen activation concentration in each region of interest was not correlated with correctness on the spatial memory task in the novice group at both map difficulties (*p* > 0.05), and the expert group only showed a high correlation with correctness in the complex map condition for the right dorsolateral prefrontal (R-DLPFC) (r = 0.792) and a moderate correlation with correctness for the right ventral lateral prefrontal (R-VLPFC) (r = 0.792). This indicates that after long-term specific training, the panel showed some right hemisphere processing advantage for the spatial memory task of oriented movements.

## 4. Discussion

### 4.1. Spatial Memory Analysis of Non-Orienteering Scenes

The purpose of this study was to compare the differences in spatial memory abilities between experts and novices in non-orienteering special scenarios. This study revealed that no significant differences in spatial memory ability emerged between experts and novices. Similar conclusions were drawn in a comparative study of basketball players and the general population who concluded that expert and non-expert athletes did not differ in their memory abilities, but rather had a superior ability to recognize motor patterns in specialized scenarios. A study by Abernethy, Baker, and Cote [31] also found that the recall performance of an expert with only one year of basketball experience was the best, whereas an expert with 10 years of basketball experience had the worst recall scores. Accordingly, the researchers concluded that there was no significant relationship between basketball experience and recall performance. It has been largely agreed that good athletes have better recall and recollection abilities, which are mainly due to the expert’s extensive specific knowledge and number of blocks, and are the result of deliberate training rather than being innate.

### 4.2. Spatial Memory Analysis of Orienteering Scenes

#### 4.2.1. Behavioral Data Analysis of Spatial Memory to Athletes

It was found that expert athletes had higher correctness rates and shorter reaction times. This suggests that, as a result of years of training, expert athletes have improved their cognitive abilities for specialized skills, i.e., improved perceptual sensitivity and effective memory strategies that enrich memory and thus make perceptual processes more adapted to the evaluation criteria needed to make categorization judgments, leading to an increase in their memory capacity for more information processing [32]. According to Ericsson and Kintsch [33] “memory is specialized skill, and this skill is developed gradually by experts in the formation of skills”, which could adequately explain the significantly greater spatial memory capacity of experts than novices in orienteering, precisely due to years of uninterrupted training of specialized skills by expert group practitioners, resulting in the acquisition of a long-term working memory capacity. Long-term specialized knowledge and skill training regularizes the structure and sequence of individual information processing variables, which, after long-term specialized training, can increase decision speed and improve decision accuracy [34].

The effect of cognitive load on memory for map recognition is one of the important issues of interest to researchers. The stimulus materials used in the study were all orienteering competition maps. Complicated maps involve more topographical information than simple maps, and complex maps are highly generalized and integrated graphics, including complex and different map shapes, symbols, and notes. Complex maps undoubtedly test the orienteers’ spatial memory ability, not only in terms of reaction time but also in terms of correctness. This study found that as the difficulty of the maps increased, both the novice group and the expert group had significantly lower correctness rates and significantly longer reaction times. This is also consistent with the study of Bethell and Shepard [35], who concluded that subjects performed cognitive processing and that reaction times were prolonged along with the complexity of the images. When subjects perform map-recognition spatial memory, they go through perceptual processes, mental image processes, memory processes, and thinking processes [36]; they and also need to correct for the map difficulty under the time pressure set by the experiment, and the complex shapes in the complex maps make the athletes spend longer on thinking and judging whether they are correct or not, thus consuming more time.

#### 4.2.2. Brain Activation Analysis of Spatial Memory in Orienteering Athletes

The study used a multichannel fNIRS system to explore changes in cerebral blood oxygen concentration in VLPFC and DLPFC of oriented athletes during a map-recognition spatial memory task [37], with the aim of understanding brain activation characteristics of athletes at different levels of the map-recognition spatial memory task. From the fNIRS results, it was learned that athletes at different levels showed differences in cerebral blood oxygen activation, with significantly lower Oxy-Hb concentrations in the expert group than in the novice group. This is also consistent with the hypothesis of the present study and the results of studies in related domains; for instance, previous studies on neural efficiency have shown that higher cognitive performance in related domains (higher cognitive performance) is negatively correlated with lower activation of the cerebral cortex, indicating that the brain’s neural resources are conserved and automated, demonstrating high cortical neural efficiency [38]. Several researchers have obtained similar results in logical reasoning [39], processing speed [40], working memory [41], problem-solving [42], driving [43], and other task processing, explaining the brain function mechanisms underlying individual performance differences. Several exercise-related studies have shown that long-term learning or training can improve the neural efficiency of the brain. The reason may be related to the optimization of cortical function and reduction in brain activation through exercise, thus explaining the in-brain functional mechanisms within the cognitive advantage of expert athletes and further validating the neural efficiency hypothesis [38,43].

In addition, differences in cerebral blood oxygenation under conditions of difficulty variables were the focus of this study, and behavioral data suggest that increased difficulty constrains individual behavioral performance. Through NIR data analysis, we found a significant increase in Oxy-Hb concentration with increasing difficulty. This suggests that cognitive load induces the intensity of prefrontal brain functional activity, that high levels of Oxy-Hb in the brain during map recognition imply a more adequate blood supply, bringing energy substances needed for brain metabolism, and that complex topography under complex map conditions mobilizes more cognitive engagement. In other studies, it has also been found that the PFC (prefrontal cortex) is activated in conflicting tasks or when the task process is complex, or the integration demands increase [44,45]. The PFC region serves as the main functional area for the cognitive generation of spatial attention, scene recognition, and spatial localization. Therefore, it fits that in the present study, athletes needed to mobilize more cognitive resources in this brain region for attention allocation under the complex map task compared to the simple map task condition.

The effect of task difficulty on cerebral blood oxygen content was also the focus of this study. The results of the present study showed that the L-DLPFC brain region had significantly lower cerebral blood oxygen activation in the novice group than in the expert only under complex map conditions. This phenomenon may be due to map difficulty changes, the novice group having less orienteering expertise and less experience in comparing field feature information, and thus being more sensitive to changes in task difficulty. Moreover, the expert group’s greater orienting skills and experience are reflected in higher surrogate effects and cognitive advantages. Complex maps involve the participation of multiple cognitive components and therefore require more cognitive resources, leading to differences in the activation of interest areas [46,47]. Therefore, the degree of activation of cerebral blood oxygen indicates that difficulty constrains the behavioral performance of orienteers, and the performance differences in the results of this study are due to differences in the perception of map specialization.

## 5. Conclusions

Good spatial memory ability is the key for orienteers to keep traveling fast and searching for targets, which is the basis for completing the competition. An in-depth understanding of orienteers’ spatial memory ability is of positive significance to improving the level of special skills. This study analyzed the behavioral and brain activation characteristics of map-recognition spatial memory of athletes at different levels and learned that expert athletes showed higher task performance and lower Oxy-Hb activation than novice athletes in completing the map-recognition spatial memory task, showing some special cognitive advantages. Different map difficulty conditions produced different performance effects and brain activation changes on spatial cognitive processing in orienteering athletes. The findings obtained are useful for deepening our understanding of the cognitive value of orienteering programs and provide theoretical support for orienteering practice interventions to modulate brain cognitive function in specific groups.

## Figures and Tables

**Figure 1 brainsci-12-01561-f001:**
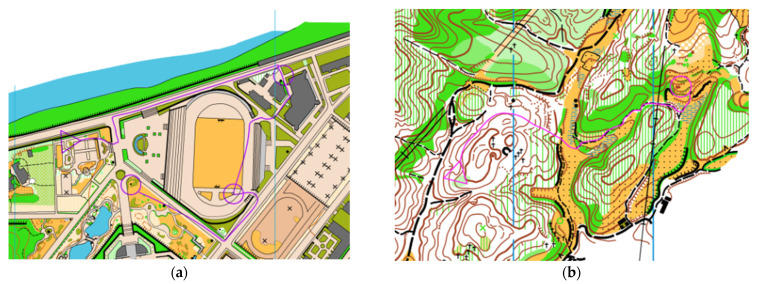
Specialized scene experiment materials. (**a**) Description of simple maps; this type of map contains mainly towns and parks as the field and the symbols of the types of features are mostly simple landforms and man-made features. (**b**) Description of complex maps; this type of map contains mainly mountain scenes in the field, and the map is mostly symbols of landforms.

**Figure 2 brainsci-12-01561-f002:**
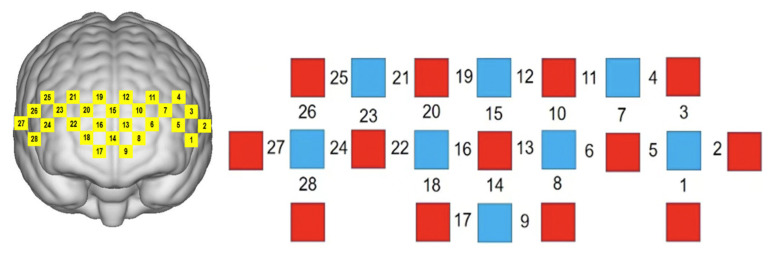
Portable fNIRS device wearing and channel layout. The yellow numbers on the left represent the corresponding detection positions of the photopolar cap in the prefrontal cortical area, the red squares on the right indicate the emitter (light source), the blue squares indicate the detector (probe), and the black numbers indicate the established channels.

**Figure 3 brainsci-12-01561-f003:**
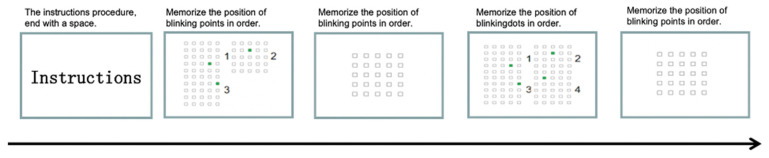
Experimental flow chart of the spatial working memory task in non-orienteering scenarios. 1. the guidance instructions before the start of the experiment, press “space” end; 2. according to the screen prompts for non-directional scenes experimental year: the location of the flash set in order of memory (3 flash points); 3. according to the memory click the corresponding luminous point position; 4. screen prompts for non-directional scenes experimental year: memorize the location of the flashing set in order (4 flashing points).

**Figure 4 brainsci-12-01561-f004:**
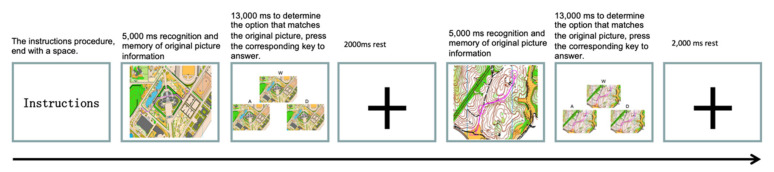
Experimental flow chart of the spatial working memory task in orienteering scenarios.

**Figure 5 brainsci-12-01561-f005:**
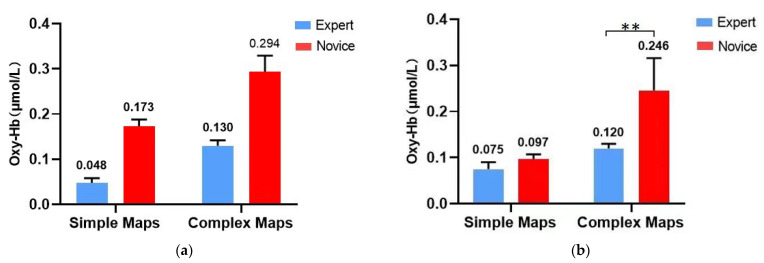
Results of fNIRS for the map-recognition spatial memory task; (**a**) L-VLPFC; (**b**) L-DLPFC; (**c**) R-VLPFC; (**d**) R-DLPFC. ** represents 0.001 < *p* < 0.01.

**Figure 6 brainsci-12-01561-f006:**
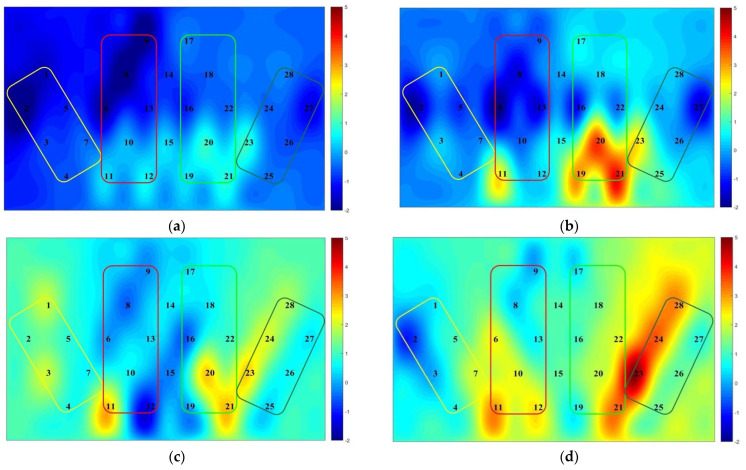
Results of fNIRS for the map-recognition spatial memory task. This figure depicts the activation level of brain regions at different maps (task difficulty) for the expert and novice groups. The more the colors are skewed toward warm colors (orange, red) the higher the activation level of the region, and the more the colors are skewed toward cool colors (blue, green) the lower the activation level of the region. Numbers represent different brain channels (**a**) Expert group simple map; (**b**) novice group simple map; (**c**) expert group complex map; (**d**) novice group complex map. The colored boxes from left to right in the picture represent different brain regions, yellow: L-VLPFC; red: L-DLPFC; grass green: R-DLPFC; olive green: R-VLPFC.

**Figure 7 brainsci-12-01561-f007:**
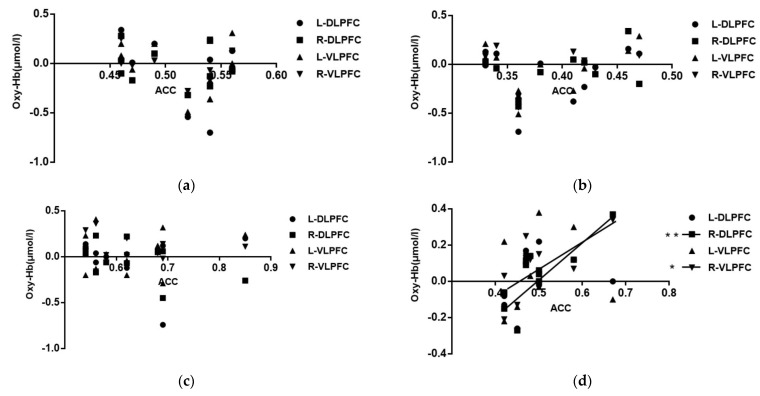
Correlation between Oxy-Hb concentration in the region of interest and the correctness rate (Note: * represents 0.01 < *p* < 0.05, ** represents *p* < 0.01). (**a**) Novice group simple map; (**b**) novice group complex map; (**c**) expert group simple map; (**d**) expert group complex map.

**Table 1 brainsci-12-01561-t001:** Demographic Information Tables.

Group	Height	Weight	BMI	Age
Novice Group	170.4 ± 6.60	65.2 ± 11.93	22.31 ± 2.80	22.33 ± 1.63
Expert Group	170.3 ± 8.00	62.3 ± 14.01	21.24 ± 3.01	21.40 ± 2.06

**Table 2 brainsci-12-01561-t002:** Correspondence table between channel layout and brain regions of portable fNIRS devices.

Brain Regions	Corresponding Channels
Left ventral lateral prefrontal cortical area (L-VLPFC)	Ch1, Ch2, Ch3, Ch4, Ch5, Ch7
Left dorsolateral prefrontal cortical area (L-DLPFC)	Ch6, Ch8, Ch9, Ch10, Ch11, Ch12, Ch13
Right ventral lateral prefrontal cortical area (R-VLPFC)	Ch23, Ch24, Ch25, Ch26, Ch27, Ch28
Right dorsolateral prefrontal cortical area (R-DLPFC)	Ch16, Ch17, Ch18, Ch19, Ch20, Ch21, Ch22

**Table 3 brainsci-12-01561-t003:** Results of spatial working memory ability for map-recognition in non-orienteering scenes (M ± SD).

	Expert Group	Novice Group
Spatial memory breadth score	5.97 ± 0.09	5.65 ± 0.08

**Table 4 brainsci-12-01561-t004:** List of behavioral results of the spatial working memory task for map recognition in directed scenarios (M ± SD).

	Novice Group		Expert Group	
Map Difficulty	Simple Map	Complex Map	Simple Map	COMPLEX MAP
Correctness rate	0.50 ± 0.03 **	0.39 ± 0.03 **	0.61 ± 0.03 **	0.48 ± 0.03 **
Reaction time	10,832.85 ± 252.97 **	11,470.71 ± 252.05 **	9218.46 ± 431.94 **	9605.31 ± 312.54 **

Note: ** represents 0.001 < *p* < 0.01.

**Table 5 brainsci-12-01561-t005:** Summary of oxygenated hemoglobin on ROI (M ± SD).

	Novice Group		Expert Group	
ROI	Simple Map	Complex Map	Simple Map	Complex Map
L-VLPFC	0.173 ± 0.016	0.294 ± 0.031	0.048 ± 0.011	0.130 ± 0.013
L-DLPFC	0.097 ± 0.011	0.246 ± 0.074	0.075 ± 0.017	0.120 ± 0.014
R-VLPFC	0.198 ± 0.011	0.417 ± 0.0348	0.054 ± 0.023	0.114 ± 0.033
R-DLPFC	0.177 ± 0.011	0.317 ± 0.020	0.025 ± 0.010	0.134 ± 0.018

**Table 6 brainsci-12-01561-t006:** Correlation results between fNIRS and behavior (Pearson correlation coefficient r).

Area of Interest	Group	Type of Map	
		Simple Map	Complex Map
L-DLPFC	Expert Group	−0.032	0.271
	Novice Group	0.259	−0.296
R-DLPFC	Expert Group	−0.483	0.792 **
	Novice Group	0.252	−0.071
L-VLPFC	Expert Group	0.178	0.047
	Novice Group	0.266	−0.396
R-VLPFC	Expert Group	−0.145	0.657 *
	Novice Group	0.317	0.059

Note: * represents 0.01 < *p* < 0.05; ** represents 0.001 < *p* < 0.01.

## Data Availability

All the data is contained within the article.

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
