# Peer review of "A Characterization of Brain Area Activation in Orienteers with Different Map-Recognition Memory Ability Task Levels—Based on fNIRS Evidence"

_brainsci, 2022, doi:10.3390/brainsci12111561_

Round 1
Author Response
Hello!
I am very glad to receive your comments, which will be very helpful for improving and perfecting the content of our paper. Thank you very much for your comments on various aspects of the language structure, English word choice, and the addition of specialized content to the article. There were some minor errors, and we appreciate your pointing them out so that we can revise and improve our research in time. We have made the following changes to a few of your comments, and we will continue to communicate with you if there are any remaining issues. We hope that we can improve our research and let more people see it.

Reviewer 2 Report
The manuscript entitled “A characterization of brain area activation in orientated athletes with different level of task in map-recognition memory ability-based on fNIRS evidence” by Yang and coworkers investigated behavioral performance and changes in cerebral blood oxygen concentration in orienteering athletes with different task difficulty and cognitive load using functional near-infrared spectroscopic imaging (fNIRS).
The authors reported that the experts have a specific cognitive advantage in map-recognition memory, showing high task performance and low cerebral blood oxygen activation than novice athletes. The correct rate in the expert group in the complex task was closely related to the activation of the right hemisphere whereas there was no significant difference in the left hemisphere.
On the whole, this study seems well conducted, methods are sound, data are convincing and the conclusions are in general supported by the data. However, there are a few issues, which need to be addressed:
· In the introduction first there should be an expansion in the text, then only an abbreviation, this applies to the words: fNIRS, Oxy-Hb
· In the Methods section a more detailed specification of the participants should be given, such as age, gender, BMI
· In table 2, first there is an expert group in the rest of the novice group, you can mistakenly analyze the results by looking at the tables, please standardize
· In Figure 5 the significance is indicated, this is not found in the description of the figure
· In the Discussion, starting at line 371, non-ending citation brackets are opened
· Some of the sentences in the discussion are too long and therefore hardly understandable for the reader, please correct the sentences in the discussion, lines:425-433; 433-435; 437-441.
Author Response
Hello!
We are very glad to receive your revision, it is very helpful to improve and perfect the content of our paper, we have made the following changes to several of your comments, if there are any remaining problems we continue to communicate, we very much hope that we can improve our research and let more people see it, thank you.

Round 2
Reviewer 2 Report
The authors responded to all my comments. I have no more comments on the manuscript.